# Overnutrition, Nasopharyngeal Pathogenic Bacteria and Proinflammatory Cytokines in Infants with Viral Lower Respiratory Tract Infections

**DOI:** 10.3390/ijerph19148781

**Published:** 2022-07-19

**Authors:** Guisselle Arias-Bravo, Gustavo Valderrama, Jaime Inostroza, Cecilia Tapia, Daniela Toro-Ascuy, Octavio Ramilo, Paz Orellana, Nicolás Cifuentes-Muñoz, Francisco Zorondo-Rodríguez, Asunción Mejias, Loreto F. Fuenzalida

**Affiliations:** 1Grupo de Virología, Instituto de Ciencias Biomédicas, Facultad de Ciencias de la Salud, Universidad Autónoma de Chile, Santiago 8910060, Chile; guisselle.arias.b@gmail.com (G.A.-B.); daniela.toro@uautonoma.cl (D.T.-A.); paz.orellana@uautonoma.cl (P.O.); nicolas.cifuentes01@uautonoma.cl (N.C.-M.); 2Urgencia Materno-Infantil, Clínica Dávila, Santiago 8431657, Chile; gvalderrama@davila.cl; 3Centro Jeffrey Modell para Diagnóstico e Investigación en Inmunodeficiencias Primarias, Centro de Excelencia en Medicina Traslacional, Facultad de Medicina, Universidad de La Frontera, Temuco 4811230, Chile; jaime.inostroza@gmail.com; 4Dirección Médica, Clínica Dávila, Santiago 8431657, Chile; cecilia.tapia@davila.cl; 5Department of Pediatrics, Division of Infectious Diseases, Nationwide Children’s Hospital—The Ohio State University, Columbus, OH 43205, USA; octavio.ramilo@nationwidechildrens.org (O.R.); asuncion.mejias@nationwidechildrens.org (A.M.); 6Departamento de Gestión Agraria, Facultad Tecnológica, Universidad de Santiago de Chile, Santiago 9170125, Chile; francisco.zorondo@usach.cl

**Keywords:** children, overnutrition, nasopharynx, pathogenic bacteria, viral respiratory infection, co-detection

## Abstract

Background: Little is known about the interaction between the nasopharyngeal bacterial profile and the nutritional status in children. In this study, our main goal was to evaluate the associations between overnutrition and the presence of four potentially pathogenic bacteria in the nasopharynx of infants with viral lower respiratory tract infections (LRTI). In addition, we determined whether changes in the nasopharyngeal bacterial profile were associated with mucosal and serum proinflammatory cytokines and with clinical disease severity. Methods: We enrolled 116 children less than 2 years old hospitalized for viral LRTI during two consecutive respiratory seasons (May 2016 to August 2017); their nutritional status was assessed, and nasopharyngeal and blood samples were obtained. *S. aureus*, *S. pneumoniae*, *H. influenzae*, *M. catarrhalis*, and respiratory viruses were identified in nasopharyngeal samples by qPCR. Cytokine concentrations were measured in nasopharyngeal and blood samples. Disease severity was assessed by the length of hospitalization and oxygen therapy. Results: Nasopharyngeal pathogenic bacteria were identified in 96.6% of the enrolled children, and 80% of them tested positive for two or more bacteria. The presence and loads of *M. catarrhalis* was higher (*p* = 0.001 and *p* = 0.022, respectively) in children with overnutrition (*n* = 47) compared with those with normal weights (*n* = 69). In addition, the detection of >2 bacteria was more frequent in children with overnutrition compared to those with normal weight (*p* = 0.02). Multivariate regression models showed that the presence and loads of *S. pneumoniae* and *M. catarrhalis* were associated with higher concentrations of IL-6 in plasma and TNF-**α** in mucosal samples in children with overnutrition. Conclusions: The nasopharyngeal profile of young children with overnutrition was characterized by an over representation of pathogenic bacteria and proinflammatory cytokines.

## 1. Introduction

Studies have shown that viral coinfections in children hospitalized for lower respiratory tract infections (LRTI) are more common in children with overnutrition, suggesting that overnutrition could influence the respiratory viral profile identified in the nasopharynx of these children [1]. In adults, the mouth, nose, and lung microbiota along with the serum proinflammatory cytokine profiles have proven to be altered in those with obesity [2].

The upper respiratory tract is colonized with different microorganisms that can potentially become pathogenic and cause secondary infections. The most frequent bacterial pathogens in children’s upper respiratory tract are *Streptococcus pneumoniae*, *Staphylococcus aureus*, *Hemophilus influenzae*, and *Moraxella catarrhalis* [3,4,5]. In addition, respiratory viral infections have proven to favor the colonization and replication of opportunistic bacteria that can lead to secondary infections, such as otitis media or pneumonia [6,7]. Respiratory syncytial virus (RSV) is the foremost cause of LRTI, leading to hospitalization in infants and young children worldwide [8,9]. The interactions between RSV and the nasopharyngeal microbiota could modulate the host immune response, potentially affecting clinical disease severity [3,4]. Whether overnutrition modulates the presence of potential pathogenic bacteria and of cytokine concentrations during viral LRTI in infants and young children is not fully known.

In this study, our main goal was to define the association between overnutrition and four potentially pathogenic bacteria in the upper respiratory tract of children with viral LRTI. In addition, we determined if changes in the nasopharyngeal bacterial profiles were associated with mucosal and serum proinflammatory cytokines and with clinical disease severity as assessed by the length of stay and duration of supplemental oxygen administration.

## 2. Materials and Methods

### 2.1. Study Population

We included 116 previously healthy infants and young children hospitalized with viral LRTI that were enrolled as part of a cross-sectional study [1] conducted during two consecutive respiratory seasons (from May 2016 to August 2017). Children were included if they were <24 months old and hospitalized for viral LRTI, including bronchiolitis, bronchitis, or pneumonia, at two medical centers in Santiago, Chile: Urgencia Materno-Infantil at Clínica Dávila and Dr. Exequiel González Cortés Hospital. Diagnoses were made based on dyspnea, signs of lower respiratory tract infections (wheezing, retractions), and/or a positive chest X-ray (infiltrates, atelectasis, and air trapping). Children were excluded if they were: (i) undernourished, (ii) neonates < 28 days old, (iii) premature, defined as a gestational age of <37 weeks, or if they had (iv) bronchopulmonary dysplasia, (v) congenital heart disease, or (vi) any previous respiratory disease, including common cold and acute otitis media, (vii) use of systemic corticosteroids within 72 h of sample collection, (viii) confirmed bacterial infections, or (ix) if they had a negative viral respiratory molecular panel at hospital admission.

Upon enrollment, a nasopharyngeal swab and blood sample were collected to classify children into three groups according to their nutritional status using the WHO Anthro 2011 v.3.2.2 program: normal weight, overweight and obese. The nutritional status was determined by z-scores according to the following anthropometric indicators: weight-for-age z-scores, length, or height-for-age z-scores, and weight-for-height z-scores. Normal weight was defined as −0.9 to 0.9 SD, overweight as 1.0 to ≤2.0 SD, and obese as >2 SD. Overnutrition (ON) was defined as ≥1.0 SD and included overweight and obese children. Undernourished children and those with nutritional risk were defined with a weight-for-age z-score and weight-for-height z-score of <1 SD below the mean and were, therefore, excluded from the study.

Disease severity was assessed using different criteria as described in other studies, including: duration of supplemental oxygen, need for mechanical ventilation, and length of hospitalization [10,11,12].

This study was approved by the ethics committees of all participating centers (Clínica Dávila, Dr. Exequiel González Cortés Hospital, and Universidad Autónoma de Chile). Written informed consent from parents or guardians was obtained before study enrollment.

### 2.2. Clinical Samples

Nasopharyngeal aspirates (NPA) were collected from all patients during the first 24 h of hospitalization, most within the first 3 h of admission. Briefly, both nostrils were aspirated without flushing using a soft catheter placed in a collection trap with 3 mL of sterile saline solution and immediately transported on ice to the laboratory. Aliquots of 1 mL were stored at −80 °C for subsequent viral, bacterial, and cytokine analyses.

In addition, blood samples (~2 mL) were collected as part of the study coinciding with standard care blood draws. Briefly, ~2 mL of blood was collected and placed in sodium heparin collection tubes (BD Vacutainer). Blood samples were then centrifuged at 1000 *g* for 15 min at room temperature and plasma divided into aliquots and stored at −80 °C.

### 2.3. Viral Analysis

Viral RNA and DNA were simultaneously extracted using 150 μL of the NPA sample using the viral RNA isolation kit NucleoSpin (Macherey-Nagel^®^, Düren, Germany) following the manufacturer’s instructions and stored at −80 °C until use. Respiratory viruses were detected using a real-time PCR kit ARGENE^®^ (bioMérieux, Marcy-I’Étoile, France) that included: respiratory syncytial virus (RSV), human metapneumovirus (HMPV), parainfluenza virus 1 to 4 (HPIV), human coronavirus (HCoV) (229E, NL63, HKU1, and OC43), adenovirus (AdV), bocavirus (HboV), influenza A (FluA), influenza B (FluB), and rhinovirus/enterovirus (HRV/HEV) [13] following the manufacturer’s instructions. Viral coinfections were defined as the presence of two or more respiratory viruses in the same sample.

### 2.4. Bacterial Identification

Aliquots of NPA aspirates stored at −80 °C were used to identify and quantitate four pathogenic bacteria (*Streptococcus pneumoniae*, *Staphylococcus aureus*, *Moraxella catarrhalis*, and *Hemophilus influenzae*) by real time (RT)-PCR with published primers [4]. Bacterial loads were measured in copies/mL and log_10_ transformed for analyses. These bacteria were selected based on previous studies demonstrating their relevant role in acute and long-term respiratory morbidity in children [12,14,15,16].

### 2.5. Cytokine Determination

Concentrations of IL-6, IL-13, IFN-γ, and TNF-**α** in NPA and plasma were measured using Magnetic Luminex^®^ assay (R&D) (R&D Systems, Inc., Minneapolis, MN, USA), according to the manufacturer’s instructions. All determinations were performed using Luminex xMAP technology (Merck Millipore) at the Institute of Biomedical Science’s Department of Virology (ICBM, Universidad de Chile).

### 2.6. Statistical Analysis

Descriptive analyses, medians (ranges), and frequency distributions were used to summarize the demographic and baseline characteristics. Data were analyzed with chi-square or Kruskal–Wallis tests, followed by Dunn’s test using a Bonferroni correction to adjust for multiple comparisons. The relative risk (RR) of increased length of hospitalization and duration of supplemental oxygen was calculated by taking the estimated Poisson regression coefficient (β) for each variable and transforming it into the eβ (exp*confidence interval) of each independent variable. The RR for mechanical ventilation was estimated by the discrete change in the probability for each independent variable. All Poisson models were adjusted for gender, age, and a dummy variable that included the RSV infection alone, RSV-viral co-infections, and no RSV infection.

Ordinary least-square multivariate models were adjusted to analyze the associations among cytokine concentrations, bacterial presence, and bacterial loads according to the nutritional status. Positive values indicate direct associations between cytokine concentration (in picograms) and bacterial profiles and vice versa. For these analyses, overweight or obese children were categorized as children with overnutrition. The statistical significance was set at two-tailed *p* < 0.05 for all analyses. All analyses were performed with Stata 14.1 software (Statacorp, College Station, TX, USA).

## 3. Results

### 3.1. Patient Demographic Characteristics and Viral Type

From May 2016 to August 2017, we enrolled a total of 116 children less than 2 years old (median age 7 (1–20) months) hospitalized with a viral LRTI. Patients were stratified based on weight into three exclusive categories: normal weight (*n* = 69, 59.4%), overweight (*n* = 30; 25.9%), and obese (*n* = 17; 14.7%). Children’s demographic and clinical characteristics are depicted in Table 1. For analyses purposes, overweight and obese infants were grouped in an additional category termed “overnutrition”. The proportion of males in the overweight cohort was lower, and 84% were vaccinated, according to the Chilean vaccine schedule, with no differences between the vaccinated and unvaccinated groups.

Of the overall cohort, 56% of the children were diagnosed with viral pneumonia, 29% with bronchiolitis, and 15% with bronchitis. Based on laboratory data and clinical parameters, none of the children showed signs or symptoms compatible with a bacterial infection, and therefore, they did not receive antibiotic treatment. All of the children included in the study required supplemental oxygen for a median duration of 5 days. Fifteen children, the majority of normal weight, required mechanical ventilation. The median duration of hospitalization was 6 days irrespective of the weight group (Table 1).

Half of the cohort (50.8%; 60/116) tested positive for a single respiratory virus, whereas 39.7% (46/116) and 8.6% (10/116) tested positive for two and three respiratory viruses, respectively. RSV was the most commonly identified respiratory virus (71.6%) either alone (34.5%) or in combination with other respiratory viruses (37.1%) (Figure 1). Detection of RSV alone was more frequent in children with normal weight (24.1%) than in those with overnutrition (10.3%; Figure 1). Moreover, RSV viral loads were higher in children with normal weight (8.06 log_10_ copies/mL) as compared to overweight (6.49 log_10_ copies/mL) and obese children (5.91 log_10_ copies/mL; *p* = 0.02), as previously reported [1].

### 3.2. Bacterial Carriage, Nutritional Status, and Severity of Infection

The overall detection of any of the four potentially pathogenic bacteria evaluated in these children with confirmed viral LRTI was 96.6% (112/116); in most cases, more than one bacterium was identified. A single bacterium was identified in 19 children (16.4%), and two or more bacteria were identified in the remaining children. Specifically, in 37 (31.9%) children, two pathogenic bacteria were identified, three bacteria in 46 (39.7%) children, and all four bacteria in 8 (6.9%) children. *H. influenzae* was the most frequently identified bacteria (68.1%), followed by *S. pneumoniae* (62.9%), *M. catarrhalis* (61.2%) and *S. aureus* (34.5%). The most frequent bacterial combinations were *S. pneumoniae*/*M. catarrhalis*/*H. influenzae* in 33 children (28.4%) followed by *S. pneumoniae*/*H. influenzae* in 11 children (9.5%) (Figure 2). Similar to viruses, the detection of a single bacterium was more common in children with normal weights (13.8%) than in those with overnutrition (4.3%). No significant differences in the distribution and load of the four different bacteria were identified based on the type of viral infection that were grouped according to the presence of only RSV, RSV and other viruses, and non-RSV viral infections.

We then analyzed the presence and bacterial burden for each bacterium according to the children’s nutritional status. Only detection of *M. catarrhalis* was more frequent (*p* = 0.001) as well as *M. catarrhalis* loads (*p* = 0.022) in children with overnutrition (which included those with overweight or obese) compared with those with normal weight (Figure 3A,B).

No significant differences were found regarding the frequency of detection and bacterial burden of *S. pneumoniae*, *H. influenzae,* and *S. aureus* according to the nutritional status. However, the detection of >2 bacteria was significantly more common in children with overnutrition compared to children with normal weight (*p* = 0.020) (Figure 4).

Detection of *S. pneumonia,* as well as bacterial loads of the four potentially pathogenic bacteria was significantly but modestly associated with increased clinical disease severity. Disease severity was assessed by an increased duration of supplemental oxygen and a lengthier hospital stay regardless of nutritional status (Appendix A).

### 3.3. Cytokine Profiles According to Nasopharyngeal Bacterial Burden and Nutritional Status

We also analyzed a panel of innate immunity cytokines in plasma and nasopharyngeal samples according to nasopharyngeal bacterial detection and quantification. We constructed different multivariable models to assess the impact of bacterial detection and bacterial loads on cytokine profiles according to the nutritional status. Of all four cytokines analyzed, only plasma and mucosal IL-6 and TNF-**α** showed significant differences and are included in Table 2 and Table 3, respectively.

Specifically, detection and loads of *M. catarrhalis* were associated with higher concentrations of IL-6 in plasma and TNF-**α** in nasopharyngeal samples only in children with overnutrition (Table 2). In addition, *S. pneumoniae* detection was associated with higher concentration in plasma IL-6 only in children with overnutrition (Table 3).

## 4. Discussion

Studies conducted in adults have shown that overnutrition can influence both the diversity of the gut microbiota profiles [17,18] and the microbial communities identified in the airway [2]. Data in children are limited. In this study, we found an association between nasopharyngeal pathogenic bacteria, proinflammatory cytokines, and overnutrition in infants and young children with viral LRTI. Although numbers are small, these data suggest that overnutrition can modulate the respiratory bacterial profile in these children.

Studies indicate that the microbiota composition of the upper airway is an important determinant for the development of LRTI and could thus influence acute disease severity as well as the future development of asthma [19]. Although numbers are small, we showed that in children with viral LRTI, the nutritional status and specifically overnutrition was associated with a distinct nasopharyngeal bacterial profile and with specific mucosal and systemic proinflammatory responses. Whether these microbiota profiles and cytokine responses will be associated with increased respiratory morbidity in children with overnutrition in the long-term is not yet known but deservers further follow-up studies.

*M. catarrhalis* was the most common bacterium identified, which is consistent with previous studies that showed the predominance of *Moraxella*-dominated nasopharyngeal profiles in upper respiratory tract infections and sinusitis in children [20,21]. Interestingly, our results showed that the frequency of the detection of *M. catarrhalis* was significantly higher, as were *M. catarrhalis* bacterial loads in children with overnutrition compared to those with normal weight. In addition, we found that concentrations of proinflammatory cytokines were higher in children with overnutrition in association with the detection of *M. catarrhalis* and to a lesser extent *S. pneumoniae*, which could partially explain the baseline proinflammatory state that has been described in obese patients. In addition, obese individuals have a higher risk of developing asthma compared to individuals with a normal weight, and it is plausible that immune and microbiome changes may play a role in the increased susceptibility of obese individuals to develop asthma [22].

Studies have shown that profiles dominated by the presence of *Streptococcus*, *Haemophilus,* and *Moraxella* in the upper airway significantly increase the risk of early allergic sensitization and persistent wheezing in school-aged children, the hallmark of the asthma phenotype [23]. Thus, our observations could be relevant and helpful for the early identification of children with an increased risk for respiratory morbidity. Monitoring changes in the bacterial profiles of the upper airway could aid in early interventions aimed at modifying the respiratory microbiota, which could potentially prevent the development of LRTI and/or of asthma in the long term.

Previous studies have described the dynamics of polymicrobial transport and bacterial cooperative relationships in patients with acute respiratory infections [24,25], which we also explored in this study. We found that the simultaneous detection of *S. pneumoniae* and *H. influenzae* was the most frequent combination, which is in agreement with previous studies [24,25]. On the other hand, only two patients showed the simultaneous co-detection of *S. pneumoniae* and *S. aureus*, which has also been previously reported in young children with RSV infections [4]. The negative association between *S. pneumoniae* and *S. aureus* appears to be associated with immune-mediated inter-species inferences [24,26,27] and possibly by the nutritional status. The combination of three or four bacteria was found more frequently in children with overnutrition, suggesting the relationship between overnutrition and an increased bacterial burden in the upper respiratory tract.

A previous study conducted in children less than 2 years old hospitalized with RSV bronchiolitis showed that the colonization with Gram-negative bacteria (*M. catarrhalis* and *H. influenzae*) was associated with higher concentrations of proinflammatory cytokines in plasma and a trend towards greater disease severity [12]. Although we also found higher plasma IL-6 concentrations in children with *M. catarrhalis* detection, the association with clinical disease severity was modest, which could be partially explained by differences in the patients studied, as we included both children with RSV and non-RSV LRTI.

Our study has limitations. As expected, RSV alone or in combination with other respiratory viruses was the main pathogen identified in most children hospitalized with LRTI, which limited our ability to perform analyses of bacterial profiles based on specific respiratory viruses. In addition, we did not conduct microbiome studies but rather limited the detection of pathogenic bacteria to the most relevant pathogens that play a role in infants and young children. Our sample size was small, which limited the number and the type of analyses that we were able to perform. Nevertheless, our results were consistent using either bivariate or multivariate analyses. Further studies performing broader microbiome analyses with a larger sample size in addition to determining the relationships between microbial communities and respiratory viruses in relation to nutritional status are warranted.

## 5. Conclusions

In conclusion, this preliminary data suggest that in infants and young children with viral LRTI, the colonizing bacteria and local and systemic host pro-inflammatory responses may be intimately connected and associated with overnutrition in children. Future studies with long-term follow-up initiatives would aid in determining whether these initial changes in microbiota profiles influence the development of long-term respiratory morbidity in children.

## Figures and Tables

**Figure 1 ijerph-19-08781-f001:**
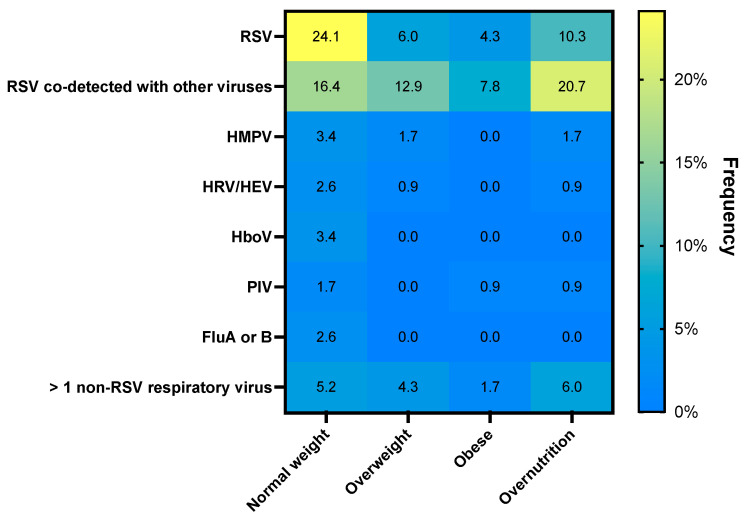
A two-dimensional heatmap showing the frequency of detection of the respiratory viruses both in mono-infection and co-infections according to patients’ nutritional status. RSV, respiratory syncytial virus; HMPV, human metapneumovirus; HRV/HEV, rhinovirus/enterovirus; HboV, bocavirus; PIV, parainfluenza virus; FluA, influenza A; FluB, influenza B.

**Figure 2 ijerph-19-08781-f002:**
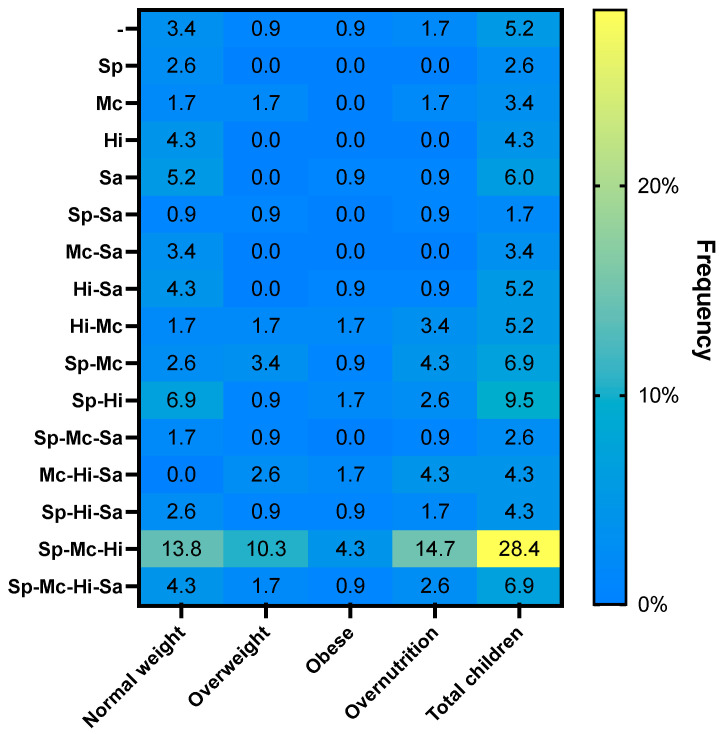
A two-dimensional heatmap showing the frequency of four nasopharyngeal pathogenic bacteria, as a single or multiple detection according to the nutritional status of children. -: no bacteria detected; Sp, *S. pneumoniae*; Mc, *M. catarrhalis*; Sa, *S. aureus*; Hi, *H. influenzae*.

**Figure 3 ijerph-19-08781-f003:**
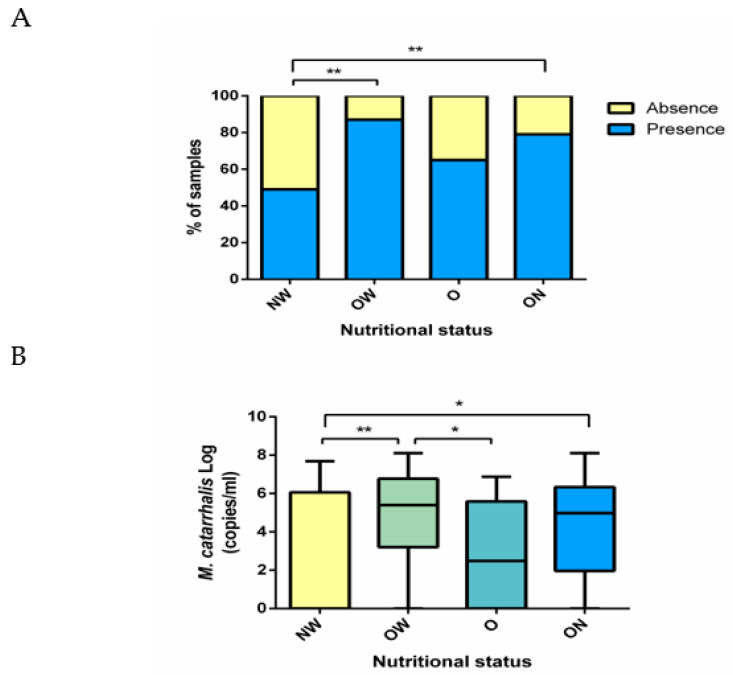
Frequency of *M. catarrhalis* detection and *M. catarrhalis* loads according to nutritional status. (**A**) Frequency of *M. catarrhalis* detection according to nutritional status. Proportions were analyzed with a chi-square test. (**B**) *M. catarrhalis* loads according to nutritional status. Kruskal–Wallis followed by Dunn’s test with Bonferroni correction were applied to adjust for multiple comparisons (*p* = 0.01). NW, normal weight; OW, overweight; O, obese; ON, overnutrition * and ** refer to significant levels at <5% and <1%, respectively. *p* < 0.05 was considered statistically significant.

**Figure 4 ijerph-19-08781-f004:**
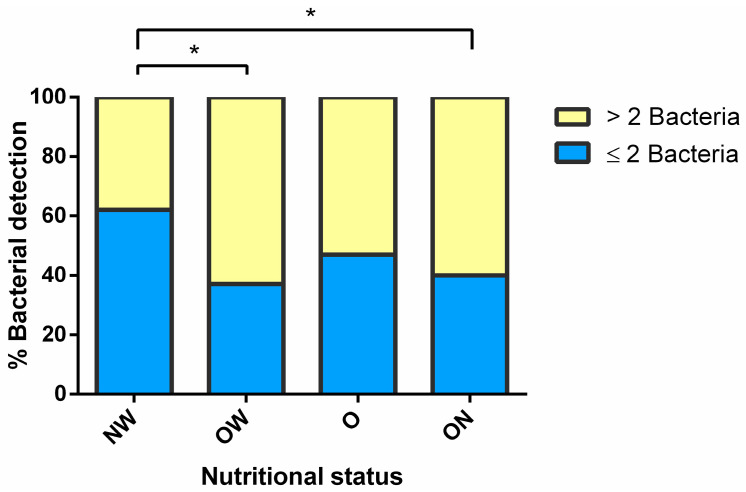
Distribution of nasopharyngeal bacterial detection according to nutritional status (*n* = 116). Proportions were analyzed by a chi-square test. *p* < 0.05 was significant (*). NW, normal weight; OW, overweight; O, obese; ON, overnutrition.

**Table 1 ijerph-19-08781-t001:** Demographic and clinical features of children according to nutritional status (*n* = 116).

Clinical Features	Number of Cases (*n* = 116)	Normal Weight (*n* = 69)	Overweight (*n* = 30)	Obese (*n* = 17)	*p*-Value
Age, months	7 (1–20)	5 (1–19)	10 (2–20)	6.5 (1–18)	0.013 ^1^
Male gender, *n* (%)	60 (51.7)	39 (56.5)	10 (33.3)	11 (64.7)	0.021 ^1^–0.030 ^2^
Breastfeeding, *n* (%)	75 (65.8)	51 (75)	14 (48.3)	10 (58.8)	0.010 ^1^
Vaccines, *n* (%)	97 (84.3)	57 (83.8)	25 (83.3)	15 (88.2)	ns
Clinical diagnosis, *n* (%)					
Pneumonia	65 (56.0)	41 (59.4)	17 (56.7)	7 (41.2)	ns
Bronchiolitis	34 (29.3)	20 (29.0)	7 (23.3)	7 (41.2)	ns
Bronchitis	17 (14.7)	8 (11.6)	6 (20.0)	3 (17.6)	ns
**Clinical Parameters**					
Days of hospitalization	6 (1–21)	6 (1–16)	6 (2–21)	7 (1–15)	ns
Days of oxygen therapy	5 (1–20)	5 (0–15)	4 (2–20)	5 (1–15)	ns
Mechanical ventilation, *n* (%)	15 (12.9)	10 (14.5)	3 (10.0)	2 (11.8)	ns

Continuous variables are expressed as medians and ranges, and categorical data as numbers and percentages (%). For continuous data, a Kruskal–Wallis test followed by a Dunn’s test, with Bonferroni correction, were conducted for multiple pairwise comparisons among groups. For categorical data, a chi-square test was performed for comparisons among groups. ^1^: Normal weight versus overweight; ^2^: Overweight versus obese. Ns, not significant.

**Table 2 ijerph-19-08781-t002:** Relationship between Nasopharyngeal Bacterial Detection and Bacterial Loads with Plasma IL-6 According to Nutritional Status.

	Overnutrition (*n* = 30)		Normal Weight (*n* = 47)	
**Bacterial detection (yes/no)**	Coefficient (95% CI)	*p*	Coefficient (95% CI)	*p*
*S. pneumoniae*	16.8 (4–29)	0.01	19.1 (−4–42)	0.10
*M. catarrhalis*	19.6 (1–38)	0.04	−2.8 (−26–20)	0.80
*H. influenzae*	−4.3(−25–17)	0.67	−0.9 (−24–22)	0.94
*S. aureus*	4.2(−12–21)	0.60	10.5 (−14–35)	0.32
**Bacterial load (log_10_ copies/mL)**				
*S. pneumoniae*	1.0 (−2.8–4.8)	0.59	1.5 (−2–5)	0.44
*M. catarrhalis*	3.9 (0–8)	0.05	1.1 (−4–6)	0.66
*H. influenzae*	0.1 (−4–4)	0.96	−2.5 (−6–1)	0.13
*S. aureus*	−0.2 (−5–4.6)	0.92	0.2 (−4–4)	0.92

Cytokine concentrations are expressed in pg/mL. Coefficients and 95% confidence intervals are reported for each covariate and analyzed using ordinary least square regressions. Analyses not yielding positive results are not included in the table (i.e., TNF-α, IL-13, and IFN-γ). *p* < 0.05 was significant. Overnutrition: obese and overweight children. Regressions were controlled for infection by VRS only, VRS coinfection, and non-VRS as a dummy variable.

**Table 3 ijerph-19-08781-t003:** Relationship between Nasopharyngeal Bacterial Detection and Bacterial Loads with Mucosal TNF-**α** According to Nutritional Status.

	Overnutrition (*n* = 39)		Normal Weight (*n* = 48)	
**Bacterial detection (yes/no)**	Coefficient (95% CI)	*p*	Coefficient (95% CI)	*p*
*S. pneumoniae*	178.3 (−26–382)	0.08	53.4 (−56–1623)	0.33
*M. catarrhalis*	131.1 (14.6–248)	0.03	90.1 (−45–226)	0.19
*H. influenzae*	−140.1 (−407–127)	0.29	78.8 (−58–216)	0.25
*S. aureus*	−25.6 (−188–137)	0.75	115.6 (−88–319)	0.25
**Bacterial load (log_10_ copies/mL)**				
*S. pneumoniae*	22.1 (−11–55)	0.19	7.4 (−14–28)	0.48
*M. catarrhalis*	43.7 (11–76)	0.01	25.5 (−2–53)	0.07
*H. influenzae*	−18.4 (−58–21)	0.34	22.7 (−9–55)	0.16
*S. aureus*	−7.8 (−39–24)	0.62	18.9 (−20–58)	0.33

Cytokine concentrations are expressed in pg/mL. Coefficients and 95% confidence intervals are reported for each covariate and analyzed using ordinary least square regressions. Regressions with no significant results are not reported in the table (i.e., IL-6, IL-13, and IFN-γ as outcome variables). *p* < 0.05 was significant. Overnutrition: obese and overweight children. Regressions were controlled for infection by VRS only, VRS coinfection, and non-VRS as a dummy variable.

## Data Availability

The datasets generated and/or analyzed during the current study are available from the corresponding authors upon reasonable request.

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
