# Peer review of "Overnutrition, Nasopharyngeal Pathogenic Bacteria and Proinflammatory Cytokines in Infants with Viral Lower Respiratory Tract Infections"

_ijerph, 2022, doi:10.3390/ijerph19148781_

Round 1

Reviewer 1 Report

The manuscript revealed the nasopharyngeal profile of certain pathogenic bacteria and viruses in children normal weight, overweight, obese and over nutrition. The data presented herein could be valuable for readers in the field. The following suggestions should be taken into consideration.

1. The authors mentioned several times that they evaluated nasopharyngeal microbiota, but in fact, only a few numbers of bacteria were investigated. Therefore, the term microbiota should be avoided so it will not be misinterpreted. 

2. The data in Tables 2 and 3 should be stratified into different weight groups and perhaps can be combined and presented as heatmaps (also with numbers and percentages) so it'd be more visual friendly and easier to compare the findings in each weight group.

3. In the Tables 4 and 5, the authors should add a column containing the hazard ratio and the actual p values, instead of the cutoff at 0.05 so the readers can access.

4. Tables 4 and 5, multivariate analyses should be performed so the association of co-infection with more than one bacterium can be determined. This would give different points of view.

Reviewer 2 Report

I read with interest the manuscript submitted for my evaluation. The authors  add knowledge between the association of respiratory infections and nutritional status.

The title is more a conclusion than a title itself.

Abstract: reflects the reason for the study. The objectives, although mentioned, should be better explained in the introduction.

Introduction: The topic is introduced correctly. Please, classify the objectives in primary and secondary. The objectives of severity and length of stay are missing.

Material and methods: Please, explain more explicitedly the exclusion criteria, some criteria are not well understood. When it says that the severity of the disease was assessed with standard criteria, could you please detail more these criteria.

Please, explain if the blood samples were obtained in isolation for the study or as part of routine analysis.

Results:

Tables 5 and 6 need clarification, because they are very confusing. Please revise the figures, the meaning of the figures, the statistical significance asterisks (appear on the normal nutrition side). Values are reported in picograms and however there are negative values. There are mistakes in the abbreviations. ON and NW do not appear in the table.

Discussion:

It is mentioned in the abstract and methods that severity of illness, length of hospital stay and oxygen therapy will be assessed. However, this information is not provided in the discussion nor in the conclusions. Limitations of the study only mention the impact of RSV but not the size sample of the study. There are few infectious cases in the overweight cohort compared to normal weight. The groups are not equal. The higher number of pneumonias compare to bronchiolitis, much more frequent at that age, is intriguing. Do you have an explanation for this?

Conclusions must be more a suggestion than a firm conclusion given the small sample size.

References: Please, follow the journal submission recommendations for the number of authors allowed per citation. 

Reviewer 3 Report

Dear Authors,

Good job putting together this important piece.

I have a few comments and suggestions -

Abstracts:

Line 26 (methods) - include the period the data referenced was collected (as you have on lines 66-67). You may also include the main statistical analysis that was used to arrive at the level of significance referenced.

Line 157: include a comma between normal weight (n=69; 59.4%) and overweight

Results:

You seem may have been interchanging chi-square test and the K-W test. Consider being consistent throughout the manuscript (e.g. Lines 213, 214, 234, etc) .

Round 2

Reviewer 1 Report

The authors have satisfactorily addressed all the concerns. However, for the heat maps in Figures 2 and 3, 'total children' should be removed so the colors could be more distinguished. 

Reviewer 2 Report

The authors have made a great effort to improve the paper.
